# Health system cost of breast cancer treatment in Addis Ababa, Ethiopia

**Tamiru Demeke**[1,2]*, **Wondimu Ayele**[1,2], **Damen Haile Mariam**[2], **Andreas Wienke**[1], **Mathewos Assefa**[3], **Adamu Addissie**[1,2], **Rafael Mikolajczyk**[1], **Susanne Unverzagt**[4☯], **Eva Johanna Kantelhardt**[1,5☯]

1 Institute of Medical Epidemiology, Biometrics, and Informatics, Interdisciplinary Center for Health Sciences, Medical Faculty of the Martin Luther University Halle-Wittenberg, Halle, Germany, 2 School of Public Health, College of Health Sciences, Addis Ababa University, Addis Ababa, Ethiopia, 3 Radiotherapy Center, Addis Ababa University, Addis Ababa, Ethiopia, 4 Institute of General Practice and Family Medicine, Interdisciplinary Center for Health Sciences, Medical Faculty of the Martin Luther Universitat Halle-Wittenberg, Halle, Germany, 5 Department of Gynaecology, Martin-Luther-University Halle-Wittenberg, Halle, Germany

☯ These authors contributed equally to this work.
* tamirudemeke@yahoo.com

**Data Availability Statement:** All relevant data are within the paper and its Supporting Information files.

## Abstract

### Background

Breast cancer is the leading cancer among women with an annual crude incidence of 27.4 per 100,000 in Ethiopia. The aims of this study were to (a) estimate the unit cost of breast cancer treatment for the standard Ethiopian patient, (b) identify the cost drivers, (c) project the total cost of breast cancer treatment for the next five years, and (d) estimate the economic burden of the disease in the main specialized tertiary hospital—Tikur Anbessa Specialized Hospital (TASH) Addis Ababa.

### Methods

Primary data were collected from health and non-health professionals. Secondary data were collected from patient's charts and official reports from various national and international organisations including data from TASH. To establish work-time estimates, we asked professionals on their time usage.

### Result

A total of US$ 33,261 was incurred to treat 52 Addis Ababa resident female breast cancer patients in TASH between July 2017 and June 2019. The unit cost of treatment for a hypothetical breast cancer patient to complete her treatment was US$ 536 for stage I and US$ 705 for stage II and III using the existing infrastructure. This cost increased to US$ 955 for stage I and US$ 1157 for stage II and III when infrastructure amortization was considered. The projected total costs of breast cancer treatment in TASH is between US$ 540,000 and US$ 1.48million. However, this will increase to US$ 870,000 and US$ 2.29 million when the existing fixed assets are changed.

**Funding:** This work was partially supported by intramural funding from Martin-Luther-University Halle-Wittenberg and a grant from the German Academic Exchange Service DAAD Project ID 57513614 received by Martin-Luther-University Halle-Wittenberg, Germany. The funders had no role in study design, data collection and analysis, decision to publish, or preparation of the manuscript.

**Competing interests:** The authors have declared that no competing interests exist.

## Conclusion

The economic burden of breast cancer treatment is high compared to the economic status of the country. Thus, it is recommended that TASH should revise its charges and breast cancer should be included in the Social and Community based health insurance scheme.

**JEL classification: H51, H75, I18, P46**

## Introduction

Breast cancer is a disease that is manifested when cells in the breast grow and divide uncontrollably resulting in a mass of tissue which is usually known as a 'tumor'. The symptoms of breast cancer are abnormal lump or swelling in the breast, the most common symptom, the lumps beside the breast or under the arm, unexplained breast pain, abnormal-nipple discharge, changes in breast texture, or changes in the skin on or around the breast [1, 2]

Thus, breast cancer is characterized by the presence of malignant tumors in one of the organ's structures, which arise from the uncontrollable reproduction of cells that have gone through a complex process of disordered transformations and may progress through direct extension or metastatic dissemination [2].

Staging describes how much cancer is present in the patient's body. The size and location of the tumor, as well as the spread the cancer to other parts of the patient's body are the factors, among others, that influence staging a breast cancer. The American Joint Committee on Cancer's TNM classification system is used to stage invasive breast cancer (AJCC) [3].

Accordingly, the basic stages of breast cancer are stage 0, I, II, III and IV and each of them are elaborated as follow.

Breast cancer Stage 0 is defined as the cancer cells are not non-invasive. It is also known as ductal carcinoma in situ. This means there is no evidence of cancer cells or non-cancerous abnormal cells breaking out of the area of the breast where they began, or of getting through to or invading neighboring normal tissue.

Breast cancer stage I refers to an extremely early stage of invasive cancer. Tumor cells have spread to normal surrounding breast tissue but are still contained in a small area at this point.

Breast cancer stage II cancer is defined as cancer that has spread beyond a specific region of the breast. It shows the number of lymph nodes that may contain cancer cells.

Breast cancer stage III breast cancer means that the cancer has spread further into the breast or that the tumor is larger than in earlier stages.

The most advanced stage of breast cancer is stage IV. It has spread to nearby lymph nodes as well as distant areas of the body outside of the breast. This means it could affect the organs, such as your lungs, liver, or brain, as well as your bones [4–7].

The exact causes of breast cancer are largely unknown but demographic change, an unhealthy lifestyle due to economic transition, urbanization, hormonal factors, a family history of breast cancer, and a sedentary lifestyle, among others, played a paramount role in the rising incidence and prevalence of breast cancer in Africa in the last two decades [8–10]. For instance, according to cancer registry reports, breast cancer incidence rose by 3.7% in Uganda and 6.5% in South Africa per year. Moreover, most of the time, African breast cancer patients visit health facilities for treatment only after their breast cancer reached advanced stage. This is mainly due to a lack of awareness about the nature of breast cancer, insufficient screening services, large distances to health facilities, low health seeking behaviour, and poverty [11]. Consequently, treatment was more often palliative than curative [12] and breast cancer is the major cause of mortality and morbidity among women in Africa [8].

The breast cancer burden in Ethiopia was compared with neighboring and peer African countries. The Global Cancer Observatory recent report indicated in 2020 female breast cancer five years prevalence was 48.52 per 100,000 which was Somalia (31.61/100,000) and South Sudan (25.72/100,000) but lower than Eritrea (48.73/100,000), Uganda (55.46/100,000), Sudan (55.62/100,000), Kenya (57.28/100,000), Nigeria (59.31/100,000), Djibouti (74.71/100,000) and South Africa (138.90/100,000) [13].

Breast cancer is the most common and frequently diagnosed disease among women in Ethiopia. According to the GLOBOCAN report, 16,133 women were newly diagnosed with breast cancer in 2020 [14]. The Five-Year National Cancer Control Plan of Ethiopia (2015–2020) indicated a need of US$ 93 million for activities related to cancer prevention, screening, diagnostic, and treatment [15]. The importance of costing has risen over time, mainly because it can help policy makers to develop appropriate health financing policies and strategies for health facilities. A costing study serves as the basis for establishing user fees, evaluating whether health care providers are cost effective, and assessing how resources are used effectively and efficiently [16–18].

This study intended to (a) estimate the total cost of breast cancer treatment for standard breast cancer patients in different stages of the disease, who completed treatment, (b) identify the most important cost drivers, (c) provide evidence on the annual budget needed for full treatment for breast cancer patients at each stage for the next 5 years, and (d) show the economic burden of the disease in Addis Ababa City.

## Methods

### Study setting

This study was undertaken in the Radiotherapy Center, Tikur Anbessa Specialised Hospital (TASH) in 2018. 84 health professionals (5 oncologists, 41 clinical oncology residents, 1 general practitioner, 3 medical physicists, 2 radiotherapy technicians, 4 pharmacists, 6 radiographers residents), 22 nurses and 6 non-health professional (1 porter, 2 chart keepers, 1 cashier, and 2 secretaries) had been working in the Center, which had a computer tomography (CT) Simulator for diagnosis, a Cobalt 60 for cancer treatment and 18 beds for cancer patients.

### Data collection

Primary data using questionnaires were collected from health professionals, who were working in the Radiotherapy Center as well as in other medical and non-medical departments of TASH from July to December 2019. The questionnaires were distributed to 22 respondents, who were selected from pathology, laboratory, pharmacy, surgery, anaesthesiology, oncology, and radiology departments. When the questionnaires were distributed to each respondents, they were asked to read the consent form and sign on it before they start to fill the questionnaire. Thus, all the respondents who filled the questionnaire signed on the consent form. Moreover, the time required to treat a single breast cancer patient was estimated by asking each professional and observing their actual time while they provide health care services.

Secondary data were collected from official reports from TASH, the Ethiopian Ministry of Health and the World Health Organisation. Other quantitative data such as employees salary, cost of pharmaceuticals, construction cost per m$^2$, Real Gross Domestic Product, foreign exchange rates, governmental total budget, and the government's budget for health and population growth were collected from official reports from TASH departments, the National Bank of Ethiopia, the Ministry of Finance, the International Monetary Fund, and the Central Statistical Agency. The tax revenues that the Ethiopian government will collect during the next five years were forcasted to obtain the total budget to be allocated for health care services.

Prior to extracting data from the patients charts, discussion was made with Oncology department head to anonymized all data to be collected from the charts. Following this, breast cancer treatment data were extracted from 55 selected patients' charts, who were treated between July 2016/17 and June 2018/19. The charts contained full information about the complete treatment given to the patients. For this study, a breast cancer patient was considered to have completed treatment if she took 8 cycles of chemotherapy.

## Methods of cost estimation

Costs of breast cancer treatment were calculated at three levels: (a) First, the unit cost of each cost driver and service, (b) the total cost for a single breast cancer patient, and (c) lastly the total cost to treat all breast cancer patients annually (2021–2025), which will be expected to be presented at TASH.

All cost drivers of breast cancer treatment were first identified, and the unit cost of each cost driver was computed using the apportioning method as indicated in Table 1. The salary data were taken from the payroll, which the human resource department of TASH provided. The salaries of human resources were estimated by multiplying hourly salaries and the total time spent to treat each patient. The existing buildings were measured in meter. The unit costs of medical and non-medical furniture and equipment were calculated using cost data from the fixed asset registration book, as well as information from contacted heads of department.

Fixed assets are defined as all types of assets that can provide service for more than one year that includes office and medical furniture, Office and medical equipment, medical and non-medical machines, buildings, land, etc.

The total costs of these items for breast cancer treatment were then calculated by multiplying the unit costs by the number of items utilized and depreciation over time. Depreciation is the value that decrease every time when a fixed asset is used. The cost of each consumable medical item was made available by the Pharmacy of the Radiotherapy center and the main

**Table 1. Sources of data and apportioning method for estimating the unit of cost of each cost driver.**

| Cost driver | Source of information | Units | Apportioning method |
|---|---|---|---|
| Salary of employees | Payroll copy from Human Resource Department | ETB converted to US$ | Average monthly salary converted to hourly wage and multiplied by the time estimated for the specific activity |
| Building | Data collected from construction professionals | Area and ETB converted to US$ | Annual depreciation divided by the total number of working hours and multiplied by the amount of time the patient received service |
| Medical and non-medical furniture and equipment | Data were collected from shops that sold the items | Number of items used | Annual depreciation divided by the total amount of working hours and multiplied by the amount of time the patient received service |
| Consumable materials and drugs | Pharmacy stores of TASH | Number of items consumed | Based on the dosage determined for the specific treatment |
| Radiotherapy | Medical physicist unit and patients register in radiotherapy room | Radiation dosis (Gray) given to patients | •The Cobalt 60 radiation source was depreciated for 5½ years and the investment divided by the total doses (Gy) provided to patients during the above-mentioned years<br>•The unit cost of the Cobalt 60 machine was depreciated by 10 years, but since the machine served for more than 10 years, the depreciation value was zero |
| Overhead costs | The costs of different materials and activities that were used jointly by the Oncology department and other (e.g. utilities, security guard) | ETB converted to US$ | 10 % of the total of the above costs were taken |

ETB: Ethiopian Birr; TASH: Tikur Anbessa Specialized Hospital; US$: United States of America Dollar.

**Table 2. Costed consumable materials.**

| Service type | Costed consumable materials | Costing method |
|---|---|---|
| Consultation | Patient card, pen, prescription papers, and gloves | Unit costs of each materials times number of materials used |
| Laboratory | Chemicals/reagents for diagnosis | Costs of each type of chemical/ reagents per mg times the quantity of chemical/ reagents used |
| Pathology | Chemicals, reagents and other related materials for diagnosis | Costs of each type of chemical/ reagents per mg times the quantity of chemical/ reagents used plus the unit costs of each materials times number of materials used |
| Ultrasound | Ultrasound paper and gel and other related materials for diagnosis | Costs of gel per mg times the quantity of gel used plus the unit costs of each materials times number of materials used |
| X-ray | Gloves and CD for diagnosis | Unit costs of gloves and CD times number of gloves and CD used |
| CT-Scan | Gloves, contrast, plastic sheet and other related materials used in the CT scan room for diagnosis | Unit costs of gloves, plastic sheet and other related materials times number of gloves, plastic and other related materials used plus costs of contrast per mg times the quantity of contrast used. |
| Anesthesia | Chemicals and other related materials for surgery. | Costs of each type of chemical/ reagents per mg times the quantity of chemical/ reagents used plus the unit costs of other related materials times number of other related materials used. |
| Surgery | Gauze, blades, and other related materials. | Unit costs of gauze, blade and other related materials times number of gauze, blade and other related materials used. |
| Radiotherapy | The Cobalt 60 radiotherapy source and related materials | Costs per Gray times the quantity of Gray plus the unit costs of other related materials times number of other related materials used. |
| Chemotherapy | Chemotherapy drugs (e.g. FEC, CMF, and Doxorubicin ACT) and other related materials. | Unit costs of each drug times the quantity of drug plus the unit costs of other related materials times number of other related materials used. |

ACT: (Adriamycin)/ Cyclophosphamide (Procytox)/ Paclitaxel (Taxol) CMF: Cyclophosphamide/Methotrexate/ Fluorouracil, FEC -Fluorouracil/Epirubicin/Cyclophosphamide

pharmacy department of TASH. The total costs of each service was obtained by multiplying the unit costs of each material by the quantity of the materials used. The grand total of all services was found by summing up the total costs of each service. To include the costs of items that are commonly utilized by all departments of TASH, 10% of the total costs of other items were taken as overhead cost.

The types of services that will be provided to breast cancer patients were identified as listed in Table 2. The costs of drugs that were prescribed and subsequently bought by patients, including endocrine treatment (Tamoxifen) were not included, because TASH does not supply these drugs.

The units and total costs of breast cancer treatment were estimated in two scenarios. The first scenario did not include depreciation of fixed assets such as medical and non-medical equipment, machines and furniture, because these items were too old and did not have book value even if they are still giving services. The second scenario includes depreciation by using the depreciation values of the fixed assets based on their current prices, if TASH replaced the existing old fixed assets by new items and renovated the existing old building. Annual deprecation for fixed assets were computed using straight-line depreciation method over different years according to the council of ministers regulation on the federal income tax [19]. Accordingly, computers, software, all medical and non-medical machines and office furniture were

depreciated by 20%, while the Cobalt 60 machine was depreciated by 15%, and building was depreciated by 5% annually. However, depreciation of the source of Cobalt 60 was calculated based on the amount of radiation Gray used and leaked every year.

The medical and non-medical consumable materials were procured between August and October 2019. To estimate the costs of the aforementioned materials in July 2017 and 2019, the costs were discounted by 15 percent due to inflation during these years and converted to US$ using the average exchange rates of the these years. Thus, the treatment costs computed for these patients reflected the costs that TASH incurred during the above-mentioned years.

Following the identification of the types of diagnosis and treatment provided, the costs incurred for breast cancer treatment for a single patient using the prevailing breast cancer treatment practice were estimated.

The new cases of breast cancer were projected based on the growth of the female population per age and the annual growth of incidence rate of breast cancer as per the record of AAPBCR.

The total costs of breast cancer treatment were calculated by multiplying the unit costs of each service by the number of breast cancer patients, who will be treated during the next five years.

These costs were estimated in two scenarios. The first scenario was using the existing fixed assets including building as they are and second scenario was after renovating/replacing the existing building and fixed assets by new ones. These costs were also estimated in three scenarios by assuming the unit costs of inputs will be increased by 10%, 15% and 20%.

It is assumed that TASH will treat all the projected new breast cancer patients using the existing practice and infrastructures. Following these steps, the total costs were estimated in Ethiopian Birr and then changed in US$ using the Wallet Investor website [20].

The economic burden of the projected cost of breast cancer treatment was estimated in terms of the share of the total health budget allocated at a national level, as it is difficult to estimate the health budget in Addis Ababa due to a lack of reliable data. Ethiopia's RGDP for the next five years was forecasted using Autoregressive and Moving Average (ARIMA) and Seasonal Auto Regressive Integrated and Moving Average (SARIMA) models [21, 22].

### Ethical consideration

The author obtained ethical approval from the Institutional Review Board of the College of Health Sciences, Addis Ababa University, prior to conducting the study. The study participants provided their consent in writing by signing on the informed consent form attached with the questionnaire.

## Results

### Costs of breast cancer treatment services for a single patient

A total of 55 breast cancer patients were assessed. Of these, three were stage I, 12 stage II, 29 stage III, eight stage IV and three of an unknown stage. These three patients were categorized as unknown stage because Oncologist did not mention the breast cancer stages in patients' charts. Because of this, these patients were excluded from further considerations. Out of 52 patients, 42 received modified radical mastectomy. When these 42 patients were disaggregated by their stage of breast cancer, 2, 8, 29 and 3 patients were at stage I, II, III and IV respectively. Similarly, among 39 patients to whom radiotherapy was provided and of these 7, 29 and 3 patients were at stage II, III and IV respectively. Chemotherapy was provided to all patients. The chemotherapy and radiotherapy treatments were provided to breast cancer patients in accordance to National Comprehensive Cancer Network Harmonized Guidelines for Sub-

**Table 3. Unit cost for treatment by breast cancer stage (April 21, 2020).**

| Type of service | Cost of treatment per stage in US$ (%) | | | | | |
|---|---|---|---|---|---|---|
| | Scenario I excluding fixed assets | | | Scenario II including fixed assets | | |
| | Stage I | Stage II | Stage III | stage I | Stage II | Stage III |
| Consultation/ Examination | 31 (60%) | 31 (4%) | 31 (4%) | 69 (7%) | 69 (6%) | 69 (6%) |
| Laboratory | 36 (7%) | 36 (5%) | 36 (5%) | 91 (10%) | 91 (8%) | 91 (8%) |
| Pathology | 20 (4%) | 20 (3%) | 20 (3%) | 72 (8%) | 72 (6%) | 72 (6%) |
| Ultrasound | 3 (1%) | 3 (0.5%) | 3 (0.5%) | 13 (1%) | 13 (1%) | 13 (1%) |
| X-Ray | 2 (0.4%) | 2 (0.3%) | 2 (0.3%) | 14 (2%) | 14 (1%) | 14 (1%) |
| CT-Scan | | 51 (7%) | 51 (7%) | | 55 (5%) | 55 (5%) |
| Surgery | 148 (28%) | 148 (21%) | 148 (21%) | 209 (22%) | 209 (18%) | 209 (18%) |
| Radiotherapy | | 118 (17%) | 118 (17%) | | 147 (13%) | 147 (13%) |
| Chemotherapy | 297 (55%) | 297 (42%) | 297 (42%) | 488 (51%) | 488 (42%) | 488 (42%) |
| Total | 536 (100%) | 705 (100%) | 705 (100%) | 955 (100%) | 1,157 (100%) | 1,157 (100%) |

Source: Authors' calculations based on the data collected from respondents

Note: Due to rounding, the sums may not be equal to the total

Saharan Africa (NCCN) guidelines by prescriping 8 cycles chemotherapy and on average 39 Gray in 13 fractions radiotherapy.

Accordingly, this study found that TASH incurred US$ 33,261 for provision of full treatment for 52 breast cancer patients. When this amount was disaggregated by breast cancer stages, TASH expended US$ 1855, 8221,19558, and 3626 to treat breast cancer patients with breast cancer stage I, II, III and IV respectively. The total costs of breast cancer treatment were again itemized by the type of diagnosis and treatment provided. Accordingly, The cost of chemotherapy, radiotherapy, laboratory and other services was $11211, 7253, 4333 and 10464 respectively.

The cost per patient for each stage was calculated and stage I, II, III and IV were found to cost US$ 618, 735,772 and 584 respectively.

## Unit cost of breast cancer treatment by stages and scenarios

Unit cost for treatment by cancer stage is presented by stages and in two scenarios (Table 3). According to scenario I (excluding the values of fixed assets), the treatment costs for breast cancer stage I was US$ 536 whereas for stage II and III the costs of treatment were US$ 705 each. These costs increased to US$ 955 to treat breast cancer stage I and US$ 1157 to treat breast cancer stage II and III each, when the values of fixed assets were included (Table 3). Chemotherapy, surgery, and radiotherapy services took the largest portions of the total cost among the services offered to breast cancer patients, as shown in Table 3.

When the replacement and renovation costs of fixed assets are included (scenario II), the costs of some services were increased while the costs of other services decreased (Table 3).

## Cost of treating breast cancer per type of input

In scenario I, consumable materials and human resources took the lion shares of the total cost of breast cancer treatment for all stages using the existing fixed assets. However, if the existing assets are replaced by the new one and the buildings are renovated, the share of each input will vary. With scenario II, the share of consumable materials and human resources was decreased to 38% and 13% for stage I and 39% and 13% for II and III respectively. The costs of medical

**Table 4. Cost of breast cancer treatment by type of input (April 21, 2020).**

| Type of inputs to be used | Treatment cost per type of input and stage of breast cancer in US$ (%) | | | | | |
| --- | --- | --- | --- | --- | --- | --- |
| | Scenario I excluding fixed assets | | | Scenario II including fixed assets | | |
| | I | II | III | I | II | III |
| Consumable materials | 360 (67%) | 456 (65%) | 456 (65%) | 360 (38%) | 456 (39%) | 456 (39%) |
| Human resource | 124 (23%) | 150 (21%) | 150 (21%) | 124 (13%) | 150 (13%) | 150 (13%) |
| Medical equipment, machines and furniture | 2 (0.5%) | 34 (5%) | 34 (5%) | 101 (11%) | 133 (12%) | 133 (12%) |
| Non-medical equipment, machines and furniture | 0.3 (0.1%) | 0.4 (0.1%) | 0.4 (0.1%) | 23 (2%) | 24 (2%) | 24 (2%) |
| Building | 0(0%) | 0.2 (0.0%) | 0.2 (0.0%) | 259 (27%) | 288 (25%) | 288 (25%) |
| Others | 49 (9%) | 64 (9.%) | 64 (9%) | 87 (9%) | 105 (9%) | 105 (9%) |
| **Total** | **536 (100%)** | **705** (100%) | **705** (100%) | **955** (100%) | **1,157 (100%)** | **1,157 (100%)** |

Source: Authors' calculations based on the data collected from respondents.

Note: Due to rounding, the sum may not equal to the total

equipment, machines and furniture increased to 11% for stage I and 12% to treat stage II and III. The costs of non-medical equipment, machines and furniture increased from 0.1% to 2% to treat stage I, II and III of breast cancer. On the other hand, the construction cost has increased from 0% to 27% for Stage I and 25% for Stage II and III (Table 4).

## Comparison of cost of treatment between current practice and NCCN guidelines

Breast cancer treatment costs for one patient at each stage of current Ethiopian practice in TASH and NCCN are similar for nearly all types of services (Table 5). Differences were observed in CT scan diagnosis, which is not prescribed in the current treatment practice while NCCN guideline prescribed it for breast cancer stage I patients. The amount of Gy prescribed in the current radiotherapy treatment practice is a little bit higher than the amount Gy prescribed in NCCN guideline. The total cost of breast cancer treatment with the current

**Table 5. Comparison of breast cancer treatment costs between current practice and NCCN guidance (as of April 21, 2020).**

| Type of service | Cost of BC treatment by stage of breast cancer (US$) | | | | | | | |
| --- | --- | --- | --- | --- | --- | --- | --- | --- |
| | As per the current practice | | | | As per NCCN Guidelines | | | |
| | Qty | I | II | III | Qty | I | II | III |
| Consultation/examination | 8 visits | 31 | 31 | 31 | 8 visits | 31 | 31 | 31 |
| Laboratory | 8 times | 36 | 36 | 36 | 8 times | 36 | 36 | 36 |
| Pathology | 1 time | 20 | 20 | 20 | 1 time | 20 | 20 | 20 |
| Ultrasound | 1 time | 3 | 3 | 3 | 1 time | 3 | 3 | 3 |
| X-Ray | 1 time | 2 | 2 | 2 | 1 time | 2 | 2 | 2 |
| CT-Scan | 1 time | | 51 | 51 | 1 time | 51 | 51 | 51 |
| Surgery | 1 time | 148 | 148 | 148 | 1 time | 148 | 148 | 148 |
| Radiotherapy | 53.68 Gy | | 118 | 118 | 50 Gy | | 160 | 160 |
| Chemotherapy | 8 Cycles | 297 | 297 | 297 | 8 Cycles | 297 | 297 | 297 |
| **Total** | | **536** | **705** | **705** | | **587** | **747** | **747** |

NCCN: National Comprehensive Cancer Network Harmonized Guidelines for Sub-Saharan Africa, Qty: Quantity

Source: Authors' computation based on the data collected from respondents and NCCN guidelines

Note: Due to rounding, the sums may not be equal to the total

**Table 6. Projection of the total costs of breast cancer treatment excluding and including the values of fixed assets from the year 2021 to 2025.**

| Year | Number of patients | Scenario I excluding fixed assets in thousands | | | Scenario I including fixed assets in thousands | | |
|---|---|---|---|---|---|---|---|
| | | 10% | 15% | 20% | 10% | 15% | 20% |
| 2021 | 670 | 540 | 560 | 590 | 870 | 910 | 950 |
| 2022 | 700 | 620 | 680 | 740 | 1,000 | 1,100 | 1,190 |
| 2023 | 723 | 700 | 800 | 910 | 1,140 | 1,300 | 1,480 |
| 2024 | 748 | 800 | 960 | 1,113 | 1,300 | 1,550 | 1,840 |
| 2025 | 776 | 910 | 1,140 | 1,410 | 1,480 | 1,850 | 2.290 |

Source: Authors' computation based on the data collected from AAPCR, CSA and data collected from TASH documents and reports

treatment practice is US$ 536 to treat a patient in stage I and US$ 705 for stage II and III compared to the total cost of breast cancer treatment per NCCN guideline as US$ 587 for stage I and US$ 747 for breast cancer stage II and III.

## Projected costs of breast cancer treatment

The number of new breast cancer patients in Addis Ababa, who might seek treatment in TASH will follow the growth of the female population per age [23] and increase from 670 in 2021 to 776 in 2025 (Table 6).

If the unit costs of materials will increase by 10 percent every year, the total cost of breast cancer treatment for TASH will increase from US$ 540,000 in 2021 to US$ 910,000 in 2025 (Table 6). If the unit costs of materials will increase by 15%, TASH will incur from US$ 560,000 in 2021 to US$ 1.14 million in 2025. However, if the unit costs of medical and non-medical materials increased by 20%, TASH will be forced to incur from US$ 590,000 in 2021 to US$ 1.41 million in 2025.

The estimated costs excluding amortization in scenario I are too low. Thus, the depreciation values of fixed assets were included assuming that the old but functional fixed assets will be either replaced by new one or renovated (scenario II). If the prices of medical and non-medical both resources will increase by 10%, the cost of breast cancer treatment will be increased to US$ 870,000 in 2021 and US$ 1.48 million in 2025 to treat 670 and 776 breast cancer patients respectively. This cost will increase to US$ 910, 000 and US$ 1.85 million in 2021 and 2025 respectively to treat the same number of breast cancer patients. When the prices of both medical and non-medical resources will increase by 20%, the cost of breast cancer treatment will increase to US$ 950, 000 and US$ 2.29 million (Table 6).

## Economic burden of breast cancer

The economic burden of the health care service is determined by the share of the total health budget as well as the gross domestic product. Accordingly, if TASH continues to provide treatment, using the existing infrastructure assuming that the costs of inputs will increaseby 10%, 15% and 20%, the share of the costs of breast cancer treatment to the total health budget will be 0.042%, 0.043% and 0.045% in 2021 respectively. These shares will be increased to 0.065%, 0.081% and 0.1% in 2025. If TASH replaces the old medical and non-medical machines, equipment and furniture by new one and the existing building is renovated and if the costs of inputs increased by 10, 15 and 20%, the share of the breast cancer to the total health budget will be 0.067, 0.070 and 0.073% in 2021 respectively and these shares will be increased to 0.105, 0.131 and 0.162% in 2025.

## Discussion

This study found considerable differences of costs for optimal treatment regimen of breast cancer patients. Chemotherapy was the most expensive, followed by radiotherapy and consultation costs. This study found that according to the current treatment practices at TASH, the total costs of breast cancer treatment significantly increased with higher stages of disease. Patients with stage two and three disease needed more expensive treatment compared to stage one. Out of the total cost incurred to optimally treat 55 breast cancer patients, 6% were spent for treatment of patients with stage I, 26% for stage II and 61% of the total cost for stage III breast cancer patients. Costs for all new breast cancer patients, who will present at TASH for treatment during the next five years were projected [24].

Guzha et al. also found somewhat similar findings that of the total cost of breast cancer treatment 6%, 47% and 35% of costs were incurred for treating stage I, II and III cancer respectively, though these costs were paid by the patients [25]. However, Nguyen et al indicated in their study that the initial treatment cost of breast cancer treatment was only US$ 128.7 for stage I but US$ 684.1 for stage III; for stage IV the treatment cost decreased to US$ 537.9. The five year total treatment cost increased from US$ 568.6 for stage I to US$ 901.8 for stage II but for stage III and IV the treatment cost decreased to US$ 816.1 and US$ 603.4 respectively. The reason for this decline was that the follow-up treatment for breast cancer in the years after the initial treatment was relatively simple [26]. Certainly the costs per patient increased with higher stages seeing the majority of patients treated for advanced stage further increased the costs. Furthermore, this study found that TASH incurred US$ 11,704 for chemotherapy which was US$ 180.66 per patient. Those studies, which were conducted at TASH, Addis Ababa, Ethiopia and Groote Schuur Hospital, Cape Town, South Africa, indicated that patients incurred far higher costs on average US$ 1,188 and US$ 1,489 for chemotherapy respectively [25]. A study conducted in Moroco on Unit price for different drugs, cost of protocols by cycle and cost of individual whole treatment, the unit cost of chemotherapy drugs for example Cyclophosphamide (1000 mg) was US$ 7.28. The total cost of chemotherapy treatment was also US$ 84.50 for AC, US$ 1105 for Docetaxel and US$ 1560 for Trastuzumab [27]. In Vietnam the average cost of chemotherapy treatment was US$ 476.48 [26]. This shows that chemotherapy costs were rather low in Addis Ababa compared to other settings were more modern substances are used.

Consumable materials took a high share out of the total costs of breast cancer treatment in all scenarios. If maintenance costs were included such as renovation of existing buildings, the share of the cost for building-renovation would be higher than the share of cost of human resources. In terms of costs of utilized for consummable inputs, the advanced stages were more costly than earlier stages of breast cancer. The costs of breast cancer treatment according to the NCCN guideline for SSA was slightly higher than the costs of breast cancer treatment according to current practice due to differences in imaging utilization and low-cost radiotherapy.

Total costs certainly depend on numbers of patients in need for therapy. The projection of the number of new breast cancer cases based on AAPBCR and CSA data indicated an increase. This projection might even be an underestimate, because with expected higher awareness, the number of Addis Ababa residents, who demand breast cancer treatment may increase. As the number of breast cancer patients will increase from year to year, the budget to be allocated would need to rise to US$ 1.5 million in 2025. This is due to the increase of investment costs for newly procured medical and non-medical equipment and furniture, as well as the cost of renovation of the existing building (increase on average by 144%). Because of this, the estimated and projected total cost of breast cancer treatment will be 1.02% and 1.251% of the total

health budget in 2021 and 2025 respectively. This will compel TASH to mobilise resources from both domestic and foreign sources. High costs for breast cancer treatment are also reported from other countries. Saber Boutayeb et al. also revealed that the cost of chemotherapy treatment for a breast cancer patient varies between US$ 507 up to US$ 30,088. In Vietnam also, the initial treatment and 5-Year total cost for treatment course was US$ 632.86 and US$ 975.01 respectvely [26]. The government of Moroco was suggested to allocate annual between US$ 13.3 million and US$ 28.6 million for breast cancer treatment [27]. These studies show that understanding the total cost of breast cancer treatment is critical, as it will inform the decision-makers on financing breast cancer treatment.

In summary, this study found that the budget required to provide breast cancer treatment for new patients depends on the number of new patients, the proportion of advanced stage as well as the detailed decision on costly targeted therapy such as trastuzumab. It should be noted that, promoting of earlier treatment of breast cancer should be given high priority to considerably reduce the economic burden of breast cancer treatment as well as increasing survival rates [28, 29].

Thus, the Ministry of Finance and Ministry of Health of Ethiopia and TASH are advised to use the data and findings of this study as a baseline, while planning and budgeting for breast cancer prevention and treatment.

Moreover, from a public health perspective, the reported data in this study can be used as a resource to develop ideas on budgeting for the different components of breast cancer therapy.

## Limitation of the study

The study has the following limitations: First, the study focused on the treatment given at TASH only. TASH is the most comprehensive cancer center in the country and sets high standard. We purposely choose a guideline concordant approach to assure the maximum benefit to the patients. Second, the study did not include the costs incurred to treat breast cancer stage IV, costs of treatment for adverse effects of medication given to breast cancer patients, and costs for breast cancer inpatient treatment. This would add additional costs but can have very high variability due to personal preferences and individualized approaches. Third, additional factors such as inflation, pandemic or difficulties of procurement may also alter prices of items.

## Conclusion

This study shows the magnitude of current costs for breast cancer service in Addis Ababa, Ethiopia. In detail, main drivers are advanced stage, investment costs such as radiotherapy machine as well as the increasing total number of patients in need of care. Hence, to alleviate the economic burden of breast cancer treatment, promoting of early diagnosis is vital. Nevertheless, it should be noted that as women's awareness about the benefits of breast cancer treatment increase, the demand for innovative breast cancer treatment would rise as well. Negotiations with pharmaceutical companies could possibly provide access to modern therapy for low-resource countries. In general, innovative financing mechanisms have to be found to meet the demand for cancer care. Hence, we recommend that international partnership should be sought to assure costly investments, policies should be carefully revised and social and community based health insurances should include breast cancer in their schemes.

## Supporting information

**S1 File. Estimted cost of breast cancer treatment at TASH.**
(XLSX)

**S2 File. Unit costs of breast cancer treatment.**
(XLSX)

**S3 File. Comparison of the cost of BC treatment.**
(XLSX)

**S4 File. Projected BC treatment cost.**
(XLSX)

## Acknowledgments

We also thank all collaborators including the employees of Tikur Anbessa Specialized Hospital, who participated and provided support in various ways for successful accomplishment of this study. Special thanks go to the oncologists and nurses who are working in the Radiotherapy Center for their major contribution to this study by providing data as required.

## Author Contributions

**Conceptualization:** Tamiru Demeke, Wondimu Ayele, Damen Haile Mariam, Andreas Wienke, Mathewos Assefa, Adamu Addissie, Rafael Mikolajczyk, Eva Johanna Kantelhardt.

**Data curation:** Tamiru Demeke.

**Formal analysis:** Tamiru Demeke, Wondimu Ayele, Damen Haile Mariam, Andreas Wienke, Adamu Addissie, Rafael Mikolajczyk, Susanne Unverzagt, Eva Johanna Kantelhardt.

**Investigation:** Tamiru Demeke, Adamu Addissie, Rafael Mikolajczyk, Eva Johanna Kantelhardt.

**Methodology:** Tamiru Demeke, Wondimu Ayele, Damen Haile Mariam, Adamu Addissie, Rafael Mikolajczyk, Susanne Unverzagt, Eva Johanna Kantelhardt.

**Project administration:** Tamiru Demeke, Eva Johanna Kantelhardt.

**Resources:** Eva Johanna Kantelhardt.

**Software:** Tamiru Demeke.

**Supervision:** Damen Haile Mariam, Andreas Wienke, Adamu Addissie, Eva Johanna Kantelhardt.

**Validation:** Rafael Mikolajczyk, Susanne Unverzagt, Eva Johanna Kantelhardt.

**Visualization:** Susanne Unverzagt, Eva Johanna Kantelhardt.

**Writing – original draft:** Tamiru Demeke.

**Writing – review & editing:** Tamiru Demeke, Wondimu Ayele, Damen Haile Mariam, Andreas Wienke, Mathewos Assefa, Adamu Addissie, Rafael Mikolajczyk, Susanne Unverzagt, Eva Johanna Kantelhardt.

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
