## [Decision Letter · Decision Letter 0]

12 Jul 2022

PONE-D-22-14104Health System cost of breast cancer treatment in Addis Ababa, EthiopiaPLOS ONE

Dear Dr. Demeke,

Thank you for submitting your manuscript to PLOS ONE. After careful consideration, we feel that it has merit but does not fully meet PLOS ONE’s publication criteria as it currently stands. Therefore, we invite you to submit a revised version of the manuscript that addresses the points raised during the review process.

Please consider all the comments.

We look forward to receiving your revised manuscript.

Kind regards,

Ahmed Mancy Mosa, Ph.D.

Academic Editor

PLOS ONE

Journal Requirements:

    "The author TD received 20% of the funding from DAAD PAGEL at Martin-Luther University University, Halle programme, Germany and 80% no specific funding for this work."

Reviewers' comments:

Reviewer's Responses to Questions

**Comments to the Author**

1. Is the manuscript technically sound, and do the data support the conclusions?

Reviewer #1: Yes

Reviewer #2: Yes

2. Has the statistical analysis been performed appropriately and rigorously? 

Reviewer #1: Yes

Reviewer #2: I Don't Know

3. Have the authors made all data underlying the findings in their manuscript fully available?

Reviewer #1: Yes

Reviewer #2: No

4. Is the manuscript presented in an intelligible fashion and written in standard English?

Reviewer #1: Yes

Reviewer #2: Yes

5. Review Comments to the Author

Reviewer #1: My suggestions:

in introduction section please refer to the main characteristics of breast cancer and define the different BC stages. It could be useful the following manuscript doi: 10.4252/wjsc.v11.i9.594 ad others.

It could be interesting compare the BC burden in Ethiopia with other countries.

If possible add the questionnaire.

It could be interesting evaluate also the costs for BC treatment in correlation with the BC survival rate related to its stage.

Please specify the meaning of fixet assets

Data reported in tables could be represented also using an aerogram.

Check accurately english form and text.

Reviewer #2: The study is an important and novel study to project future and current cost of the breast cancer treatment. Hence, early preparation for future prevention and treatment. The data is not fully available due to several restrictions; confidential issues-patient's chart etc. Methods have been explained in detail and obtained ethical approval. The authors need to consider to re-check the typing error (spelling/format) in several lines (Example in line 63, 166, 203, 230, 255-259).

6. PLOS authors have the option to publish the peer review history of their article (what does this mean?). If published, this will include your full peer review and any attached files.

Reviewer #1: No

Reviewer #2: No

---

## [Author Response · Author response to Decision Letter 0]

9 Aug 2022

Responses to reviewers

Reviewer #1: My suggestions:

in introduction section please refer to the main characteristics of breast cancer and define the different BC stages. It could be useful the following manuscript doi: 10.4252/wjsc.v11.i9.594 ad others.

My response: The suggestion is accepted and I did accordingly and the following paragraphs are added from line number 47-55.

Breast cancer is a disease that is manifested when cells in the breast grow and divide uncontrollably resulting in a mass of tissue which is usually known as a ‘tumor’. The symptoms of breast cancer are abnormal lump or swelling in the breast, the most common symptom, the lumps beside the breast or under the arm, unexplained breast pain, abnormal-nipple discharge, changes in breast texture, or changes in the skin on or around the breast(1, 2)

Thus, breast cancer is characterized by the presence of malignant tumors in one of the organ's structures, which arise from the uncontrollable reproduction of cells that have gone through a complex process of disordered transformations and may progress through direct extension or metastatic dissemination(2).

Staging describes how much cancer is present in the patient's body. The size and location of the tumor, as well as the spread the cancer to other parts of the patient’s body are the factors, among others, that influence staging a breast cancer. The American Joint Committee on Cancer's TNM classification system is used to stage invasive breast cancer (AJCC)(3). Accordingly, the basic stages of breast cancer are defined as follow;

• Breast cancer Stage 0. The disease is non-invasive. This means it hasn’t broken out of your breast ducts.

• Breast cancer Stage I. The cancer cells have spread to the nearby breast tissue.

• Breast cancer Stage II. The tumor is either smaller than 2 centimeters across and has spread to underarm lymph nodes or larger than 5 centimeters across but hasn’t spread to underarm lymph nodes. Tumors at this stage can measure anywhere between 2 to 5 centimeters across, and may or may not affect the nearby lymph nodes.

• Breast cancer Stage III. At this stage, the cancer has spread beyond the point of origin. It may have invaded nearby tissue and lymph nodes, but it hasn’t spread to distant organs. Stage III is usually referred to as locally advanced breast cancer.

• Breast cancer Stage IV. The cancer has spread to areas away from your breast, such as your bones, liver, lungs or brain. Stage IV breast cancer is also called metastatic breast cancer(4).

Reviewer #1 suggestion: 

It could be interesting compare the BC burden in Ethiopia with other countries.

My response: The suggestion is accepted and I did accordingly and the following paragraphs are added from line number 84-88.

The breast cancer burden in Ethiopia was compared with neighboring and peer African countries. The Global Cancer Observatory recent report indicated in 2020 female breast cancer five years prevalence was 48.52 per 100,000 which was Somalia (31.61/100,000) and South Sudan (25.72/100,000) but lower than Eritrea (48.73/100,000), Uganda (55.46/100,000), Sudan (55.62/100,000), Kenya (57.28/100,000), Nigeria (59.31/100,000), Djibouti (74.71/100,000) and South Africa (138.90/100,000)(5). 

Reviewer #1 suggestion: 

If possible, add the questionnaire.

My response: Yes, possible. I attached 

Reviewer #1 suggestion: 

It could be interesting evaluate also the costs for BC treatment in correlation with the BC survival rate related to its stage.

My response: The idea is nice. But I want to kindly explain that the study was focused on the cost of breast cancer (BC) treatment from the provider side if a BC patient is provided the full treatment regardless of her survival. My study takes a hypothetical woman for each stage who will be provided standard treatment and completed her treatment instead of taking the real patient. The reason is, as I mentioned earlier, to know the health system cost that will be incurred to treat a BC patient who is presented at TASH with different stages of BC. In other words, the study tries to estimate how much money Tikur Anbessa Specialized Hospital (TASH) will incur to treat a BC who will be presented at TASH with stage I, 2 and 3 regardless of the economic status of the patient as the costs that the patient will incur for treatment is not taken into account. Thus, the assumption is that if a BC who is presented at TASH with stage I and provided standard treatment and completed her treatment, as many studies indicated, the patient’s relative survival (RS) will be around 98%, if a patient is presented at TASH with breast cancer stage II, gets standard treatment and if she completed her treatment, her RS will be around 90%. If a BC patient is presented at TASH with BC stage III, gets standard treatment and if she completed her treatment, her RS will be about 70%. Therefore, it is possible to evaluate the costs of BC treatment in correlation with the RS at each stage. In fact, this evaluation will lead us the estimation of cost effectiveness of the treatment. Nevertheless, since this not the objective of our study, we did not do as you suggested. 

Reviewer #1 suggestion: 

Please specify the meaning of fixed assets

My response: I accepted the suggestion and the following statements is added from line number 143 to 145.

Fixed assets are all types of assets that can provide service for more than one year that includes office and medical furniture, Office and medical equipment, medical and non-medical machines, buildings, land, etc. 

Reviewer #1 suggestion: 

Data reported in tables could be represented also using an aerogram.

My response: I tried to draw a graph that shows the number on left axis and the percentage at right axis because I wanted to show the numbers and percentages at the same time in order to make clear the data analysis and results clear. Nevertheless, readers including co-authors suggested that the graph was confusing and it is difficult to understand. So, I was compelled to put these two figures in one table. 

Reviewer #1 suggestion: 

Check accurately english form and text.

My response: Prior to submitting to the journal, I gave to the English native speaker who was also researcher in my university and she improved it. As per your comment, I read it again repeatedly and made minor changes. 

Reviewer #2 suggestion: 

The study is an important and novel study to project future and current cost of the breast cancer treatment. Hence, early preparation for future prevention and treatment. The data is not fully available due to several restrictions; confidential issues-patient's chart etc. Methods have been explained in detail and obtained ethical approval. The authors need to consider to re-check the typing error (spelling/format) in several lines (Example in line 63, 166, 203, 230, 255-259).

My response: I did accordingly.

Moreover, when I added some ideas as per the reviewer #1 comment, the following new references are added.

References

1. Britannica-The Editors of Encyclopedia. Breast cancer. Encyclopedia Britannica Encyclopedia Britannica Inc.; 07 July 2022.

2. Rodrigo Sanches Peres, Manoel Antônio dos Santos. Breast Cancer, Poverty and Mental Health: Emotional Response to the Disease in Women from Popular Classes. Rev Latino-am Enfermagem. 2007;15:786-91.

3. Beth C. Freedman , Alyssa Gillego, Susan K. Boolbol. The Surgical Management of Invasive Breast Cancer. In: Darius S. Francescatti, Melvin J. Silverstein, editors. Breast Cancer-A New Era in Management. USA: Springer; 2014.

4. Cleveland Clinic. Breast Cancer Ohio, USA: Cleveland Clinic; n.d. [cited https://my.clevelandclinic.org/health/diseases/3986-breast-cancer 25th July 2022].

5. The Global Cancer Observatory. Cancer Today France: IARC, WHO; 2021 [updated March, 2021; cited https://gco.iarc.fr/today/home 25th July 2022].

---

## [Decision Letter · Decision Letter 1]

26 Aug 2022

PONE-D-22-14104R1Health System cost of breast cancer treatment in Addis Ababa, EthiopiaPLOS ONE

Dear Dr. Demeke,

Thank you for submitting your manuscript to PLOS ONE. After careful consideration, we feel that it has merit but does not fully meet PLOS ONE’s publication criteria as it currently stands. Therefore, we invite you to submit a revised version of the manuscript that addresses the points raised during the review process.

Please, accurately, consider the comments  . 

We look forward to receiving your revised manuscript.

Kind regards,

Ahmed Mancy Mosa, Ph.D.

Academic Editor

PLOS ONE

Journal Requirements:

Reviewers' comments:

Reviewer's Responses to Questions

**Comments to the Author**

1. If the authors have adequately addressed your comments raised in a previous round of review and you feel that this manuscript is now acceptable for publication, you may indicate that here to bypass the “Comments to the Author” section, enter your conflict of interest statement in the “Confidential to Editor” section, and submit your "Accept" recommendation.

Reviewer #1: All comments have been addressed

Reviewer #2: (No Response)

2. Is the manuscript technically sound, and do the data support the conclusions?

Reviewer #1: Yes

Reviewer #2: Yes

3. Has the statistical analysis been performed appropriately and rigorously? 

Reviewer #1: N/A

Reviewer #2: I Don't Know

4. Have the authors made all data underlying the findings in their manuscript fully available?

Reviewer #1: Yes

Reviewer #2: No

5. Is the manuscript presented in an intelligible fashion and written in standard English?

Reviewer #1: Yes

Reviewer #2: No

6. Review Comments to the Author

Reviewer #1: Authors improved the overall quality of the manuscript. Authors addressed all my comments. The manuscript is now accetable for publication in PlosOne

Reviewer #2: Thank you for addressing the previous comments. Kindly consider to convert the point form sentences in the Introduction part into paragraph. Please check any typographical errors in the document (Font color etc).

7. PLOS authors have the option to publish the peer review history of their article (what does this mean?). If published, this will include your full peer review and any attached files.

Reviewer #1: No

Reviewer #2: No

---

## [Author Response · Author response to Decision Letter 1]

29 Aug 2022

Reviewer #2: Thank you for addressing the previous comments. Kindly consider to convert the point form sentences in the Introduction part into paragraph. Please check any typographical errors in the document (Font color etc).

I changed the point form sentences in to paragraphs and make all letters in the same (black) color.

---

## [Decision Letter · Decision Letter 2]

13 Sep 2022

Health System cost of breast cancer treatment in Addis Ababa, Ethiopia

PONE-D-22-14104R2

Dear Dr. Demeke,

We’re pleased to inform you that your manuscript has been judged scientifically suitable for publication and will be formally accepted for publication once it meets all outstanding technical requirements.

Kind regards,

Ahmed Mancy Mosa, Ph.D.

Academic Editor

PLOS ONE

Additional Editor Comments (optional):

Reviewers' comments:

Reviewer's Responses to Questions

**Comments to the Author**

1. If the authors have adequately addressed your comments raised in a previous round of review and you feel that this manuscript is now acceptable for publication, you may indicate that here to bypass the “Comments to the Author” section, enter your conflict of interest statement in the “Confidential to Editor” section, and submit your "Accept" recommendation.

Reviewer #2: All comments have been addressed

2. Is the manuscript technically sound, and do the data support the conclusions?

Reviewer #2: Yes

3. Has the statistical analysis been performed appropriately and rigorously? 

Reviewer #2: N/A

4. Have the authors made all data underlying the findings in their manuscript fully available?

Reviewer #2: Yes

5. Is the manuscript presented in an intelligible fashion and written in standard English?

Reviewer #2: Yes

6. Review Comments to the Author

Reviewer #2: (No Response)

7. PLOS authors have the option to publish the peer review history of their article (what does this mean?). If published, this will include your full peer review and any attached files.

Reviewer #2: No

---

## [Editor Report · Acceptance letter]

22 Sep 2022

PONE-D-22-14104R2 

Health System cost of breast cancer treatment in Addis Ababa, Ethiopia 

Dear Dr. Demeke:

I'm pleased to inform you that your manuscript has been deemed suitable for publication in PLOS ONE. Congratulations! Your manuscript is now with our production department. 

Kind regards, 

on behalf of

Dr. Ahmed Mancy Mosa 

Academic Editor

PLOS ONE